# The Effects of Gabapentin on Post-Operative Pain and Anxiety, Morphine Consumption and Patient Satisfaction in Paediatric Patients Following the Ravitch Procedure—A Randomised, Double-Blind, Placebo-Controlled, Phase 4 Trial

**DOI:** 10.3390/jcm11164695

**Published:** 2022-08-11

**Authors:** Dariusz Fenikowski, Lucyna Tomaszek, Henryk Mazurek, Danuta Gawron, Piotr Maciejewski

**Affiliations:** 1Department of Thoracic Surgery, Institute of Tuberculosis and Lung Diseases, Rabka-Zdrój Branch, 34-700 Rabka-Zdrój, Poland; 2Faculty of Medicine and Health Sciences, Andrzej Frycz Modrzewski Krakow University, 30-705 Kraków, Poland; 3Department of Pneumonology and Cystic Fibrosis, Institute of Tuberculosis and Lung Diseases, Rabka-Zdrój Branch, 34-700 Rabka-Zdrój, Poland; 4Health Institute, State University of Applied Sciences in Nowy Sącz, 33-300 Nowy Sącz, Poland

**Keywords:** Ravitch procedure, gabapentin, morphine, postoperative pain, pain assessment method, anxiety, patient satisfaction, nursing care

## Abstract

The aim of the study was to investigate whether the use of pre- and postoperative gabapentin can decrease postoperative pain, morphine consumption, anxiety and side effects, as well as improve patient satisfaction. A total of 56 patients, 9–17 years of age, undergoing a modified Ravitch procedure, were randomised (allocation ratio 1:1) to receive multiple perioperative doses of gabapentin (preoperatively 15 mg/kg, postoperatively 7.5 mg/kg, two times per day for three days) or a placebo. All the patients received intravenous infusion of morphine, paracetamol and non-steroidal anti-inflammatory drugs. Metamizole was given as a “rescue drug”. The observation period included the day of surgery and three postoperative days. The primary outcomes were postoperative pain intensity (at rest, during deep breathing and coughing). Additional outcomes included the consumption of morphine, the total number of doses of metamizole, anxiety, postoperative side effects and patient satisfaction. Median average and maximal pain scores (on the day of surgery and on the second postoperative day) were significantly lower only in the gabapentin group at rest (*p* < 0.05). Compared to the placebo group, gabapentin treatment reduced the demand for morphine on the first postoperative day (median 0.016 vs. 0.019 mg/kg/h; *p* = 0.03) and the total number of metamizole doses (median 1 vs. 2 *p* = 0.04). Patient satisfaction was significantly greater in the gabapentin group (median 10 vs. 9; *p* = 0.018). Anxiety and postoperative side effects were similar in both groups (*p* > 0.05). Pre- and postoperative gabapentin administration as part of a multimodal analgesic regimen may decrease postoperative pain, opioid consumption and demand for a “rescue drug”, as well as improve patient satisfaction.

## 1. Introduction

The Ravitch procedure is the reconstructive operation of an anterior chest wall deformity [1]. Next to the Nuss procedure, it is considered the surgery of choice in the treatment of pectus excavatum [2], pectus carinatum and mixed pectus deformities—a combination of both pectus excavatum and carinatum [3]. In Buchwald’s modification, the method consists of cutting off a certain number of deformed costal cartilages from the sternum, shortening these cartilages and reattaching them with single, non-absorbable polyester sutures to the previously surgically modelled sternum [3]. The surgery should be optimally performed before puberty in order to avoid deterioration of the defect in the period of rapid growth. However, the choice of the most appropriate age for surgery is still a matter of debate [4].

Due to the area of operation, which is well innervated by the intercostal nerves, the procedure is one of the most painful. The pectoralis major and the rectus abdominis muscle are subject to surgical trauma. Moreover, a drain placed in the retrosternal space can cause pain [3]. The pain prevents deep breathing and coughing up of the secretions from the bronchial tree, which may result in pneumonia. Effective pain control is a prerequisite for the prevention of complications and ensuring the patient’s mental and physical well-being in the postoperative period [5].

The optimal postoperative pain management involves a multimodal approach. Multimodal analgesia regimens can utilise varying combinations of opioid and non-opioid analgesics and regional analgesic techniques [6]. Thoracic epidural, in many institutions, is the standard of perioperative analgesia in children with chest wall deformity [7,8,9]. However, intravenous analgesia may also be an effective pain management option [10].

The perioperative administration of gabapentin is an increasingly common element of multimodal therapy. The drug has been approved for the treatment of neuropathic pain in adults (e.g., in diabetes, after herpes zoster infection) and epilepsy in all age groups [11,12]. As an off-label drug, gabapentin is used in the treatment of pain after various types of surgical interventions, chiefly in adults [13], and less often in children [12]. This is mainly due to its analgesic properties, described by some authors, and the effect on reducing the need for opioids. As a result, the incidence of adverse events, such as nausea and vomiting, is reduced [14]. Among other benefits of using the drug, researchers mention its anxiolytic effect and the influence on the improvement of patient satisfaction [13]. However, there is conflicting evidence to support the use of the drug. A systematic re-view and meta-analysis of 281 randomised trials conducted in the adult population by Verret et al. [15] did not reveal clinically significant analgesic effects of gabapentinoids in their perioperative use. Similar observations were made by Egunsola et al. [12] in their systematic review covering studies of the paediatric population conducted until 2017.

With this background, we designed this trial to investigate if the use of gabapentin as a component of a multimodal analgesic regimen reduced pain scores, consumption of intravenous morphine, anxiety and side effects, as well as whether it improved patient satisfaction following the Ravitch procedure in paediatric patients.

## 2. Materials and Methods

### 2.1. Trial Design, Setting

This was a single-centre, randomised, blinded, placebo-controlled, parallel-group (allocation ratio 1:1), phase 4 trial conducted in Poland. The trial took place at the Department of Thoracic Surgery of the Institute of Tuberculosis and Lung Disease, Rabka Zdrój Branch, from May 2017 to December 2020. The trial protocol, in line with the Declaration of Helsinki, was approved by the Local Bioethics Committee at the National Institute of Tuberculosis and Lung Diseases in Warsaw (decision numbers: KB-6/2017; KB-125/2019). The trial was registered with Clinicaltrails.gov (ID: NCT03393702) and was reported in accordance with CONSORT 2010 guidelines [16].

### 2.2. Participants

Patients were eligible for enrolment in the trial if they were between 9 and 17 years of age, with an American Society of Anesthesiologists (ASA) physical classification of 1 (normal healthy patients) or 2 (patients with mild systemic disease), scheduled for an elective Ravitch procedure, in whom postoperative intravenous morphine analgesia was used, and written informed consent was obtained from the patients (from the age of 16) or their legal guardians. Exclusion criteria were lack of postoperative chest drainage, inability to rate pain, a known allergy or sensitivity to gabapentin or morphine, chronic pain or daily analgesic use and those diagnosed with psychiatric disorders or epilepsy or treated oncologically, in whom postoperative thoracic epidural analgesia was used.

### 2.3. Interventions

The patients were randomly assigned to the gabapentin (standard care + gabapentin; experimental group) or placebo group (standard care + placebo; control group). In the experimental group, oral gabapentin was supplemented one hour before surgery (in a dose of 15 mg/kg), and 2 times per day on the first, second and third postoperative day (at 6.00 a.m. and 6.00 p.m. in a dose of 7.5 mg/kg). Patients in the control group were given placebo capsules instead of gabapentin. The nursing staff administered the trial medication.

The perioperative medicine protocol was standardised in both groups. It included premedication with oral midazolam (0.2–0.5 mg/kg), intravenous antiemetic prophylaxis with ondansetron hydrochloride (0.1 mg/kg up to 4 mg), pre-emptive analgesia with paracetamol (15 mg/kg intravenous) and non-steroidal anti-inflammatory drugs (ketoprofen 1 mg/kg intravenous or ibuprofen 10 mg/kg rectally in children up to 14 years of age). Anaesthesia was induced with an injection of fentanyl (1–5 μg/kg) and propofol (3–5 mg/kg). A neuromuscular blockade was achieved with rocuronium bromide (1 mg/kg in children up to 10 years of age) or pancuronium bromide (0.1 mg/kg). Anaesthesia was maintained with propofol infusions and desfluranum in an oxygen/air mixture. Intraoperative analgesia was obtained with fentanyl (1–5 μg/kg) administered every 20–30 min.

Postoperative analgesia included continuous intravenous infusion of morphine (0.02–0.06 mg/kg/h), intravenous paracetamol administered every 6 h (15 mg/kg, a maximum of 60 mg/kg daily) and non-steroidal anti-inflammatory drugs every 8 h (in children > 14 years of age, ketoprofen intravenously—1 mg/kg, a maximum of 200 mg daily; in children < 14 years of age, ibuprofen orally or rectally—up to 30 mg/kg daily). All patients, in case of pain > 2/10, could receive a supplemental opioid (increasing the flow rate of morphine by 10–30% and/or administration of morphine in a bolus (half-hourly dose)). Metamizole was given as a “rescue drug” (20 mg/kg; a maximum of 2.5 g daily). In case of a sedation score ≥ 3, nurses could decrease the flow rate of morphine by 10–30%.

Postoperative bundle also contained intravenous antiemetics. Ondansetron hydrochloride was given every 8 h (0.1 mg/kg up to 4 mg) up to the second postoperative day. In patients who failed prophylaxis with ondansetron, metoclopramide hydrochloride (0.1–0.2 mg/kg) and/or dexamethasone (0.15 mg/kg up to 5 mg) were used according to physician decision.

Blood pressure, heart rate, respiratory rate and oxygen saturation measured by pulse oximetry were monitored continuously during the observational postoperative period (in the postoperative intensive care unit).

### 2.4. Outcomes

The primary endpoint with respect to efficacy of pain management was postoperative pain intensity. Additional outcomes included the consumption of morphine (daily and total), the total number of doses of a “rescue drug”, anxiety, postoperative side effects and patient satisfaction.

Assessment of postoperative pain utilised an 11-point Numerical Rating Scale (NRS; 0 = no pain and 10 = worst pain imaginable) according to the three-step method (Figure 1). This method, used in our department since 2012, obliged nurses to carry out pain assessment under dynamic conditions (i.e., at rest, during deep breathing and coughing). Pain-relieving interventions were according to the physician’s orders and the guidelines available in the department. On the day of surgery (PD 0) and on the first day after surgery, nurses measured pain for the first 4 h, every hour, then at least every 4 h. On the second and third day after surgery, pain was assessed at least 4 times a day. Furthermore, pain was evaluated 30 min after an additional analgesic was given. The average pain score and maximal pain score (at rest, during deep breathing and coughing) were calculated for each postoperative day in all patients.

The State-Trait Anxiety Inventory for adolescents (STAI) and children between 9 and 14 years of age (STAI-C) was used to measure state and trait anxiety [17,18]. State anxiety is defined as a temporary reaction to adverse events (e.g., hospitalisation for surgery), whereas trait anxiety represents a fairly stable characteristic related to personality. The STAI and STAI-C scores range from 20 to 80 and 20 to 60, respectively. The obtained raw values were expressed on the sten scale (5–6 sten = a moderate level of anxiety; ≥7 sten = a high level of anxiety). State anxiety was assessed one day prior to surgery and on the third postoperative day. Trait anxiety was measured once before surgery (together with state anxiety).

Analgesia-related adverse effects, including nausea and vomiting, urinary retention, pruritus, dizziness, oxygen desaturation (defined as a fall in oxygen saturation to lower than 94% for at least 4 min), bradycardia (a heart rate 20% lower than the baseline value), a sedation score ≥ 3 points and others, were documented. Sedation levels were controlled together with pain using a 5-point scale as follows: 1 = patient anxious, agitated; 2 = patient cooperative, oriented, tranquil; 3 = patient asleep, easy to wake up; 4 = patient asleep, difficult to wake up; 5 = patient asleep, does not respond to painful stimulus. If excessive sedation occurred, the nurses could reduce or stop the flow rate of morphine.

Patient satisfaction with their postoperative analgesia was obtained after the end of the trial. Satisfaction was evaluated on an 11-item scale: 0 = very dissatisfied, 10 = very satisfied.

### 2.5. Sample Size

Pain intensity was the primary endpoint measure used to evaluate the efficacy of gabapentin in our trial. The minimal sample size was calculated based on Rusy at al.’s study [19], who showed that pain intensity scores in paediatric spinal fusion patients in the recovery room were significantly lower in the gabapentin group than in the placebo group (2.5 vs. 6.0; *p* < 0.001). Assuming alpha of 0.01 and the same standard deviation for mean pain scores 2.6, at least 19 patients in each group were needed to obtain 90% power of difference detection.

### 2.6. Randomisation

The clinical trial nurse and physician enrolled patients in the intervention (Figure 2). The clinical trial nurse was responsible for informing patients and their parents about the intervention, patients’ rights and nursing care in the perioperative period. The nurse also taught children how to correctly assess the intensity of pain and instructed them to complete the State-Trait Anxiety Inventory. The main task of the physician was to obtain the patient’s/parent’s consent and write an order for gabapentin (the dose was based on body weight). An independent hospital pharmacist assigned patients to interventions according to a computer-generated randomisation list. The pharmacist dispensed either gabapentin or a placebo. The capsules, packed in an envelope marked with “gabapentin” and the personal identity number code of the patient, were identical in appearance. For safety reasons, the pharmacist placed the information about whether the patient received gabapentin or placebo in a second sealed envelope, which was stored in a designated place. The envelope could be opened only in the case of life-threatening adverse events; otherwise, it was to be returned intact after the end of treatment.

### 2.7. Blinding

The children and their parents, nursing staff, surgeons, anaesthesiologists, investigators and data analysts were blinded to group assignments. It should be noted that the principal investigator performed both the duties of the study director and was responsible for data analysis.

### 2.8. Statistical Methods

Statistical analysis was performed using STATISTICA v.13.3 (TIBCO Software Inc. (2017), Kraków, Poland). Normality of data distribution was tested with Shapiro–Wilk tests—only body height, body weight and diastolic blood pressure had normal distribution. The Mann–Whitney U test and Wilcoxon probability tests were used to compare the differences between the two groups (independent and dependent, respectively) when variables were not normally distributed. Otherwise, the Student t test was used. In order to maintain homogeneity in the presentation of data, all quantitative variables were expressed as a median and upper and lower quartile. The chi-square test or Fisher exact test was used for testing relationships among categorical variables—the data were reported as absolute numbers and percentages. The two-sided significance of tests was *p* < 0.05 for all analyses. Glass’s delta or Cohen’s d were reported as measures of effect size (comparison of groups with equal size). Effect size estimates were interpreted as small (0.2 to <0.5), medium (0.5 to <0.8) and large effects (≥0.8) [20].

## 3. Results

The trial involved 56 patients, between 9 and 17 years of age, undergoing surgical chest wall reconstruction due to pectus excavatum (*n* = 47; 83.9%), pectus carinatum (*n* = 8; 14.3%) or mixed pectus deformities—a combination of both pectus excavatum and carinatum (*n* = 1; 1.8%). The two groups were compared for demographic and clinical data, in which no significant differences were observed (Table 1). Most of the subjects were boys (89.3%) and patients with ASA 1 (92.8%). The median age of the patients was 14, body weight was 54 kg, and body height was 171 cm. Minimum and maximum duration of surgery was 95 min and 230 min, respectively, whereas the duration of anaesthesia ranged from 140 to 325 min. The median time of drainage was 48 h.

There were no significant differences between the groups regarding intraoperative fentanyl administration (median 0.3 [0.2; 0.4] mg vs. 0.4 [0.3; 0.5] mg; Z = −1.61; *p* = 0.12) and bolus of morphine given after extubation within the operation theatre (median 1 [0; 2] mg vs. 0 [0; 1.5] mg; Z = 0.78; *p* = 0.43).

### 3.1. Primary Outcome—Postoperative Pain

Postoperative pain scores at rest, during deep breathing and coughing according to NRS are shown in Table 2. The study revealed that patients receiving gabapentin felt significantly less pain, both average (on the day of surgery) and maximal (on the day of surgery and on the second postoperative day), at rest. Gabapentin had a large effect only on maximal pain scores on the day of surgery (Glass’s delta = 0.8)—in other cases, the effect size was medium (Glass’s delta in the range of 0.5–0.7).

A moderate positive correlation was found between median average pain at rest on postoperative days 0–3 and the number of pain measurements (R = 0.38; t = 3.02; *p* = 0.004). During this period, nurses took pain measurements more frequently in the placebo than control group (median 32 [29; 35] vs. 29 [27; 31]; Z = −3.02; *p* = 0.002; Glass’s delta = 0.7 = medium effect size).

### 3.2. Additional Outcomes

#### 3.2.1. Morphine Consumption

Table 3 shows the average consumption of morphine. The median amount of morphine used was significantly lower in the gabapentin group than in the placebo group only on the first postoperative day (21 vs. 25 mg; Z = −2.18; *p* = 0.03). Demand for morphine in the gabapentin group was 0.016 mg/kg/h, while in the placebo group, this was 0.019 mg/kg/h. Gabapentin had a medium effect on morphine consumption (Glass’s delta = 0.5).

Patients treated with gabapentin (postoperative days 0–3) required less morphine modification by bolus administration and an increased/decreased flow rate compared to the placebo group (median 5 [3; 7] vs. 6 [4; 11]; Z = −2.46; *p* = 0.01). An absolute value of Glass’s delta of 0.7 shows a medium effect size.

One patient from the gabapentin group and two patients who received the placebo were given an additional opioid (tramadol 50 mg) on the third postoperative day.

#### 3.2.2. Metamizole Consumption

There was a significant difference between the gabapentin and placebo groups in terms of the number of doses of metamizole (median 1 [0; 2] vs. 2 [1; 4]; Z = −2.03; *p* = 0.04). An absolute value of Glass’s delta of 0.7 shows a medium effect size.

#### 3.2.3. Anxiety

The Mann–Whitney test results indicated no significant difference between the gabapentin and placebo groups in terms of preoperative and postoperative anxiety state (*p* > 0.05). Moderate levels of preoperative state anxiety were recognised in 46.4% of the total patients (*n* = 26), whereas high anxiety was felt by 37.5% of the patients (*n* = 21). Analysis of the correlation demonstrated statistically significant high associations between preoperative and postoperative state anxiety (R = 0.6, t = 6.19, *p* < 0.0001).

The Wilcoxon test showed significant differences in children’s preoperative and postoperative anxiety (Figure 3). Compared to preoperative anxiety, postoperative anxiety scores were significantly reduced both in the gabapentin group (median 6 [5; 6] vs. 5.5 [4; 6]; Z = 2.68; *p* = 0.007) and the placebo group (median 7 [5; 7] vs. 5.5 [3.5; 6]; Z = 3.65; *p* = 0.0002). Both results had medium effect sizes (Cohen’s d = 0.7).

#### 3.2.4. Analgesia-Related Adverse Effects

Side effects of the analgesic treatment used occurred with a similar frequency in both groups (Table 4). At least one incidence of oxygen desaturation and a sedation score of 3 were registered in 78.6% and 46.4% of patients, respectively. A moderate positive correlation was found between the total episodes of oxygen desaturation and the total hours of oxygen supplementation in the postoperative period (R = 0.53; t = 4.59; *p* < 0.0001). The analysis demonstrated a statistically significant weak positive correlation between oxygen supplementation and morphine use (R = 0.27; t = 2.09; *p* = 0.04). The median oxygen supplementation time was significantly lower in the gabapentin group than in the placebo group (8 [0–11] vs. 13 [7–17] h; Z = −2.35; *p* = 0.02).

Common side effects during the postoperative period were nausea and vomiting (50%), despite antiemetic administration. No statistically meaningful differences between the gabapentin and placebo groups were observed in the frequency of antiemetics used (*p* > 0.05)—the median number of doses for all patients was 7 [6; 9]. Apart from this, we found urinary retention in 46.4% of patients, but only three patients required bladder catheterisation due to insufficient pharmacological provocation. There were no significant differences in postoperative haemodynamic variables (mean blood pressure, heart rate, respiratory rate and oxygen saturation) (*p* > 0.05).

#### 3.2.5. Patient Satisfaction

Patient satisfaction with their postoperative pain management was high (minimum 7, maximum 10) and was significantly greater in the gabapentin group compared to the placebo group: median 10 [9: 10] vs. 9 [8; 10]; 2.37; Z = 2.22; *p* = 0.018. An absolute value of Glass’s delta of 0.5 suggests a medium effect size.

## 4. Discussion

The findings of this trial indicate that the perioperative use of oral gabapentin as a component of a multimodal pain management protocol may reduce postoperative pain at rest, consumption of morphine and demand for a “rescue drug”, as well as improve patient satisfaction following the Ravitch procedure in paediatric patients. There was no difference in anxiety or in the side effect profile between the gabapentin group and the placebo group.

Both the present trial and previous trials [19,21] found that a multiple-dose regimen of gabapentin may decrease both pain intensity and demand for opioids. This trial showed that administering gabapentin 15 mg/kg before surgery and 7.5 mg/kg every 12 h for 3 days after surgery resulted in a reduction in pain scores at rest (on the day of surgery and on the second postoperative day) and the need for morphine on the first postoperative day. Similar findings, for the immediate postoperative period through day 2, were reported in paediatric spinal fusion patients with idiopathic scoliosis by Rusy et al. [19]. It is worth noting that these patients received the same daily dose of gabapentin both pre- and postoperatively as in our trial, but in the 5-day postoperative period, the total daily dose was divided into three single doses. The recent randomised, double-blind study conducted by Anderson et al. [21] also confirmed the beneficial effects of gabapentin only through the first 48 h after surgery, despite administering a higher maintenance dose of this medication to patients (10 mg/kg every 8 h for 5 days).

In our institution, proper postoperative pain relief is our priority. The median average pain scores were <1/10, whereas, in adolescents after scoliosis surgery, in the study by Mayell et al. [22], this was 2–7/10 at rest and 4–7/10 during coughing or movement. We believe that adequate pain relief was achieved thanks to the use of preemptive analgesia [23,24], the combination of non-opioid and opioid analgesics, known as multimodal analgesia in the postoperative period [25], regular pain measurements [26] according to the three-step method, i.e., at rest, during deep breathing and coughing [9] and adequate supervision of the implementation of pain relief procedures in clinical practice [27]. However, the effect of gabapentin on pain scores was medium, with the exception of maximum pain on the day of surgery, which was large. Grøvle et al. [28] draw attention to the need to analyse the use of rescue medication when interpreting the efficacy of an active drug, as this may lead to an underestimation of its effect. The researchers noted that in trials demonstrating a small or a medium effect size of the investigational drug, subjects receiving an active drug consumed 17–30% less “rescue drug” than those receiving a placebo. The link between gabapentin and the number of doses of metamizole as a “rescue drug” was visible in our trial. Administration of a placebo was associated with a significantly greater number of doses of metamizole. We used metamizole as a “rescue drug” due to its strong analgesic effect (the analgesic strength of a 2.5 g dose of metamizole is comparable to a 10 mg dose of morphine; level II according to EBM) and synergistic action with non-steroidal anti-inflammatory drugs, paracetamol and opioid analgesics [29]. Short-term metamizole use for the treatment of postoperative pain in children seems to be well tolerated and safe (no clinical signs of agranulocytosis were reported) [30].

One objective of this trial was to evaluate satisfaction with the pain control provided to the paediatric patients, which is being used increasingly as an indicator of quality of care. Past studies looking at gabapentin associated with postoperative satisfaction have yielded mixed results. Doleman et al. [13], in a study on adult patients, showed that perioperative oral gabapentin increased patient satisfaction. These findings are in line with our results and those by Salman et al. [31], who reported parents’ satisfaction after sevoflurane anaesthesia in their children undergoing an adenoidectomy or tonsillectomy. On the other hand, Anderson et al. [21] did not find a connection between gabapentin and parental satisfaction. Both our patients and the parents in the above-mentioned study were very satisfied with the postoperative analgesia.

Anxiety is the most common emotion experienced by patients scheduled for surgery [32]. Over 80% of our patients suffered from moderate to severe preoperative anxiety. These trial findings showed that gabapentin had similar anxiolytic effects to the placebo—all patients felt less postoperative anxiety in comparison to their preoperative state. We hypothesised that the intravenous infusion of morphine eliminated the anxiolytic effects of gabapentin, which was observed in thoracic paediatric patients in whom postoperative pain management was treated by epidural infusion of 0.2% ropivacaine with fentanyl 5.0 μg/mL [33]. It is worth emphasising that this trial and the above-mentioned one were conducted according to the same protocol in terms of administering gabapentin, non-opioid drugs or rescue medications. The reduction in anxiety in the morphine administration period was confirmed in a recent animal trial [34].

### Strengths and Limitations

The major strength of this trial is that it is randomised and double-blind. The trial was developed according to CONSORT guidelines. However, the trial, being single-centre and conducted chiefly among boys (the majority of patients were male patients), limits the possibility of generalising our results to all paediatric patients. The study was also limited by the small sample size and subjective nature of pain assessment. Findings related to patient satisfaction may be limited due to the lack of measurement using validated tools.

## 5. Conclusions

Gabapentin, used as a component of a multimodal analgesic regimen, had a medium effect size on reducing pain intensity at rest, on the consumption of morphine during the early postoperative period and on the demand for a “rescue drug”, as well as on improving patient satisfaction, following the Ravitch procedure in paediatric patients.

## 6. Practical Implications of the Trial

Taking into account the medium effect size of gabapentin on the above-mentioned postoperative variables and the similar profile of the side effects to a placebo, we believe that gabapentin may be used as a part of a multimodal analgesic regimen, though only in patients with a higher level of pain than those in our study. Postoperative administration of gabapentin should be limited to the second postoperative day, as the benefits of its administration are not apparent beyond this period. Our observations are consistent with the results of Anderson et al.’s research [21]. We agree with Fabritius et al. [35] that firm evidence for the use of gabapentin is still lacking. Therefore, larger randomised trials are needed to confirm the benefits of gabapentin in controlling acute pain after thoracic surgery.

## Figures and Tables

**Figure 1 jcm-11-04695-f001:**
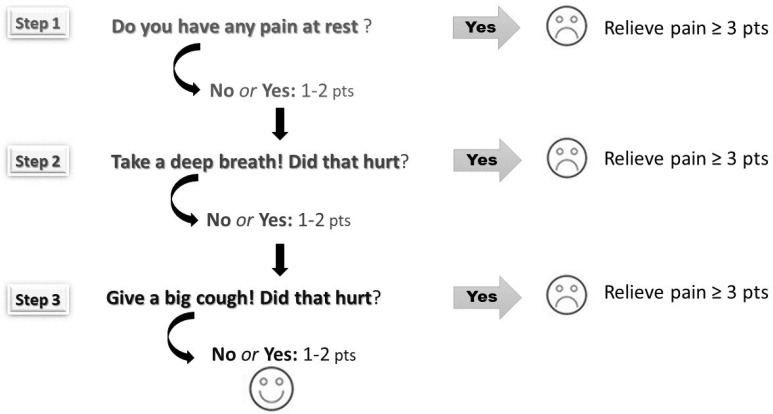
The three-step pain assessment method (pain was assessed at rest, during deep breathing and coughing according to Numerical Rating Scale ranging from 0 points to 10 points).

**Figure 2 jcm-11-04695-f002:**
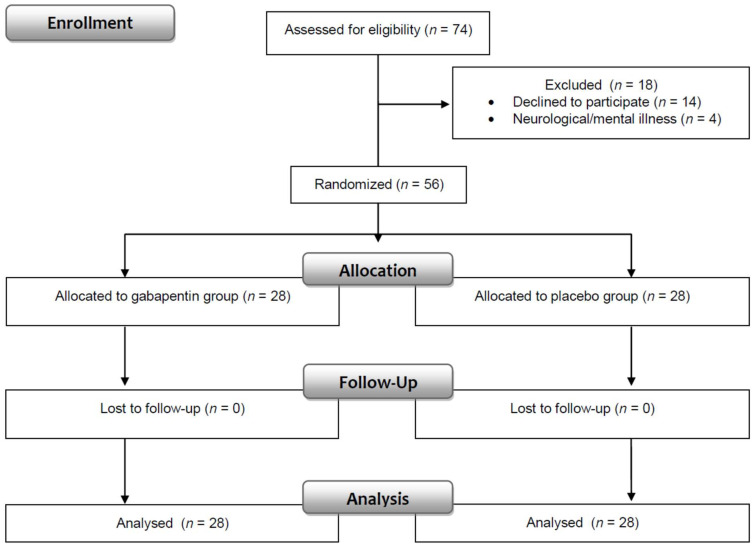
Consort flow diagram.

**Figure 3 jcm-11-04695-f003:**
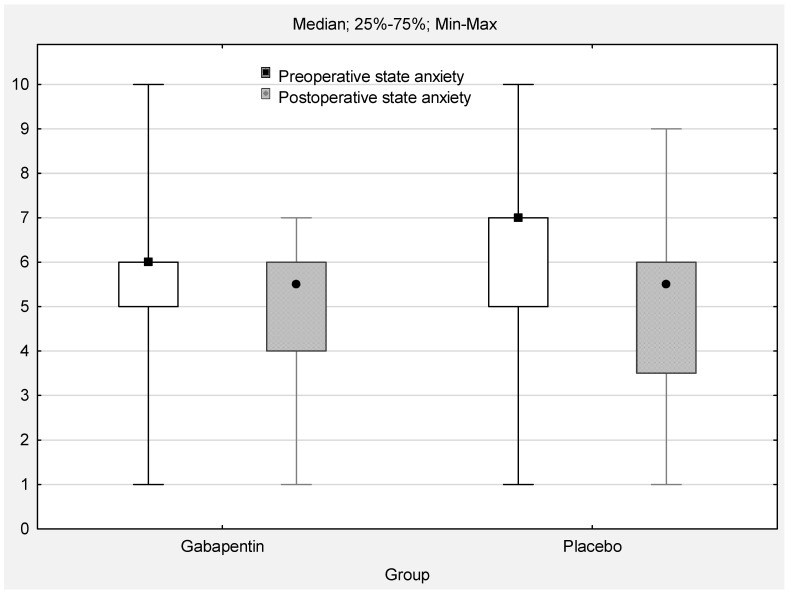
Pre- and postoperative state anxiety (sten; range 0–10; *p* < 0.05).

**Table 1 jcm-11-04695-t001:** Demographic and clinical data in the gabapentin and placebo groups.

Variable		Gabapentin *n* = 28	Placebo *n* = 28	*p* Value
Age (years)		14 [13; 15]	15 [13; 16]	0.32
Body height (cm)		172 [167; 176]	168 [161; 177]	0.47
Body weight (kg)		54 [45; 59]	54 [46; 61]	0.86
BMI		18 [16; 20]	19 [18; 20]	0.34
Sex	Girls	1 (3.6)	5 (17.9)	0.19
	Boys	27 (96.4)	23 (82.1)	
Trait anxiety	(sten)	4.5 [3.5; 5]	5 [3.5; 6]	0.33
Before induction	Heart rate (beat min^−1^)	88 [76; 95]	84 [78; 89]	0.25
	Systolic blood pressure (mmHg)	115 [106; 125]	120 [103; 127]	0.79
	Diastolic blood pressure (mmHg)	70 [60; 77]	70 [60; 75]	0.81
	Oxygen saturation (%)	98 [97; 99]	99 [98; 99]	0.38
ASA	1	26 (92.9)	26 (92.9)	1.00
	2	2 (7.1)	2 (7.1)	
Duration of anaesthesia (min)		197 [180; 219]	193 [176; 203]	0.588
Duration of surgery (min)		135 [121;159]	140 [128; 149]	0.75
Duration of extubating (min)		15 [13; 25]	15 [10; 20]	0.42
Intravenous morphine (hour)		80 [76; 90]	82 [74; 89]	0.72
Drainage (hour)		48 [45; 69]	47 [44; 66]	0.11

Results presented as median, lower and upper quartile or absolute number (percentage); the gabapentin and placebo groups were comparable with respect to demographic and clinical data (*p* > 0.05); BMI— Body Mass Index; ASA—American Society of Anesthesiologists physical status.

**Table 2 jcm-11-04695-t002:** Postoperative pain scores at rest, during deep breathing and coughing according to NRS in the gabapentin group (*n* = 28) and the placebo group (*n* = 28).

PD	Pain	Average Pain		*p* Value	Maximal Pain		*p* Value
		Gabapentin	Placebo		Gabapentin	Placebo	
0	At rest	0.3 [0.1; 0.8]	0.8 [0.3; 1.1]	0.049 **	3.0 [1.0; 4.0]	4.0 [3.0; 5.0]	0.02 ***
1		0.0 [0.0; 0.0]	0.0 [0.0; 0.3]	0.38	0.0 [0.0; 0.0]	0.0 [0.0; 2.0]	0.56
2		0.0 [0.0; 0.0]	0.0 [0.0; 0.2]	0.08	0.0 [0.0; 0.0]	0.0 [0.0; 1.5]	0.04 *
3		0.0 [0.0; 0.0]	0.5 [0.0; 0.3]	0.23	0.0 [0.0; 0.0]	0.0 [0.0; 1.5]	0.3
0	During deep breathing	0.3 [0.0; 0.6]	0.5 [0.05; 0.7]	0.33	2.0 [0.0; 2.5]	2.0 [0.5; 3.0]	0.37
1		0.0 [0.0; 0.2]	0.0 [0.0; 0.3]	0.42	0.0 [0.0; 1.0]	0.0 [0.0; 2.0]	0.24
2		0.0 [0.0; 0.2]	0.0 [0.0; 0.3]	0.64	0.0 [0.0; 1.0]	0.0 [0.0; 1.0]	0.63
3		0.0 [0.0; 0.0]	0.0 [0.0; 0.0]	0.37	0.0 [0.0; 0.0]	0.0 [0.0; 0.0]	0.51
0	During coughing	0.5 [0.0; 0.9]	0.5 [0.2; 0.8]	0.62	2.0 [0.0; 3.0]	2.0 [1.0; 3.0]	0.65
1		0.0 [0.0; 0.3]	0.3 [0.0; 0.6]	0.14	0.0 [0.0; 2.0]	1.5 [0.0; 2.0]	0.06
2		0.1 [0.0; 0.3]	0.3 [0.0; 0.5]	0.29	0.0 [0.0; 2.0]	1.0 [0.0; 2.0]	0.30
3		0.0 [0.0; 0.0]	0.0 [0.0; 0.1]	0.17	0.0 [0.0; 0.0]	0.0 [0.0; 0.0]	0.42

PD—postoperative day; NRS—Numerical Rating Scale, range 0–10; results presented as median, lower and upper quartile; medium effect size: Glass’s delta = 0.5 *, Glass’s delta = 0.6 **; large effect size: Glass’s delta = 0.8 ***.

**Table 3 jcm-11-04695-t003:** Morphine consumption (mg) in the gabapentin group (*n* = 28) and the placebo group (*n* = 28).

Postoperative Day	Gabapentin	Placebo	*p* Value
0	34 [29; 37]	36 [31; 48]	0.17
1	21 [19; 24]	25 [21; 32]	0.03 *
2	15 [12; 19]	19 [14; 22]	0.09
3	5 [2; 9]	5 [2; 9]	0.96
0–3	18 [16; 21]	21 [18; 28]	0.10

Results presented as median, lower and upper quartile; Glass’s delta = 0.5 * (medium effect size).

**Table 4 jcm-11-04695-t004:** Incidence of side effects in the gabapentin group (*n* = 28) and the placebo group (*n* = 28) in the observation period (PD 0–PD 3).

Variable	Gabapentin	Placebo	*p* Value
^1^ Oxygen desaturation < 94%	19 (67.9)	25 (89.3)	0.05
^2^ Nausea and vomiting	13 (46.4)	15 (53.6)	0.59
^3^ Incidence of sedation score of 3	14 (50.0)	12 (42.9)	0.59
^4^ Urinary retention—pharmacological provocation	10 (35.7)	9 (32.1)	0.77
Pruritus	1 (3.6)	1 (3.6)	1.00
Bradycardia	1 (3.6)	1 (3.6)	1.00
Dizziness	0 (0.0)	1 (3.6)	1.00

Results presented as absolute number and percentage; ^1^ total incidence of oxygen desaturation: *n* = 165; ^2^ total incidence of vomiting: 1 incidence (*n* = 13), 2 incidences (*n* = 8), 3 incidences (*n* = 1), 4 incidences (*n* = 4), 7 incidences (*n* = 2); vomiting: on the day of surgery—32% (*n* = 18), on the first postoperative day—30.3% (*n* = 17), on the second postoperative day—12.5% (*n* = 7), on the third postoperative day—12% (5/40); ^3^ incidence of sedation score of 3 only on the day of surgery (*n* = 26; 46.4%); ^4^ pharmacological provocation was ineffective—urinary retention was treated by bladder catheterisation in 1 patient (3.6%) in the gabapentin group and 2 patients (7.1%) in the placebo group.

## Data Availability

A dataset will be made available upon request to the corresponding authors one year after the publication of this study. The request must include a statistical analysis plan.

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
