# Peer review of "The Effects of Gabapentin on Post-Operative Pain and Anxiety, Morphine Consumption and Patient Satisfaction in Paediatric Patients Following the Ravitch Procedure—A Randomised, Double-Blind, Placebo-Controlled, Phase 4 Trial"

_jcm, 2022, doi:10.3390/jcm11164695_

Round 1

Answers to Reviewer 1

Response 1: There are several methodologic reasons to include a placebo-controlled group as opposed to an active control group. First, the use of a placebo group in a double-blind, randomized, controlled trial is the most rigorous test of treatment efficacy for evaluating a medical therapy. Second, placebo-controlled trials can be conducted with fewer patients than active control trials. This is because trials with a placebo group offer the opportunity to compare outcomes under conditions in which there is maximal ‚treatment separation‛ (group exposed to an investigational treatment vs. group not exposed to the same investigational treatment), increasing the likelihood of detecting beneficial and/or harmful treatment-related effects (i.e., increased statistical power). This has ethical implications because fewer subjects are potentially exposed to toxic or ineffective treatments. Third, a placebo group can be used as an ‚add on‛ to standard of care in comparison to an investigational treatment added to standard of care. Therefore, the true added benefit (or risk) of the new therapy could be evaluated. ...Last, placebo-controlled trials are essential in the selection of trial endpoints when subjective measures ..... This is especially true for studies involving pain relief..'' [Castro M. (2007). Placebo versus best-available-therapy control group in clinical trials for pharmacologic therapies: which is better?. Proceedings of the American Thoracic Society, 4(7), 570–573]. Taking into account the above considerations, we cannot agree that a placebo is the biggest limitation of this study. Please note that in our study true comparator was not a placebo but standard care (all patients received). We compared the standard of care + gabapentin vs. standard of care + placebo. Placebo was used only for patient blinding – it was not possible to evaluate the effect of additional gabapentin treatment in another way. We have added to subsection 4.1. Strengths and Limitations: ‘’The study was also limited by small sample size and subjective nature of pain assessment. Findings related to patient satisfaction may be limited due to the lack of measurement using validated tools’’.

Response 2: We have left one primary endpoint in the revised Materials and Methods section. The primary endpoint with respect to the efficacy of pain management was postoperative pain intensity. Additional outcomes included the consumption of morphine, the total number of doses of a "rescue drug", anxiety, postoperative side effects and patient satisfaction. The minimal sample size was calculated based on pain intensity.

Response 3: We have left one primary endpoint – pain intensity. In the revised subsection 2.4. Outcomes we have specified that: · the average pain score and maximal pain score (at rest, during deep breathing and coughing) were calculated for each postoperative day in all patients; · the daily and total amount of morphine consumption was calculated.

Response 4: We have provided this information in the revised Introduction section: ‘’Multimodal analgesia regimens can utilize varying combinations of opioid, non-opioid analgesics, and regional analgesic techniques [6]. Thoracic epidural, in many institutions, is the standard of perioperative analgesia in children with chest wall deformity [7, 8, 9]. However, intravenous analgesia may also be an effective pain management option [10]

Response 5: The outcomes were mentioned in the last paragraph of the original version of the manuscript.

Response 6: The drugs were administered in accordance with the Summary of Product Characteristics in force in Poland. The intravenous ketoprofen formulation was only licensed for patients aged from 15 years. The intravenous ibuprofen was not available when the study was provided (it is licensed for patients aged from 6 years from 2021-09-24). For this reason, two different NSAIDs were used. Ibuprofen was given orally if the patient was able to swallow the drug. Drugs were dosed according to body weight, e.g. ketoprofen 1 mg/kg, but in patients weighing > 70 kg the maximum dose was 200 mg daily.

Response 7: Postoperative bundle also contained intravenous antiemetics. Ondansetron hydrochloride was given every 8 hours (0.1 mg/kg up to 4 mg) up to the second postoperative day. Only in patients who failed prophylaxis with ondansetron, metoclopramide hydrochloride (0.1-0.2 mg/kg) and/or dexamethasone (0.15 mg/kg up to 5 mg) were used according to physician decision. This information we have provided in the revised 2.3. Interventions subsection.

We did not assess the risk of nausea and vomiting for each patient according to Apfel score for adults. Knowing, that the postoperative risk of nausea and vomiting according to the Apfel scale is 39-61% (gender, use of postoperative opioids) and moderate or high according to the VPOP score for the pediatric population (age, duration of anaesthesia >45 min, and multiple opioid doses), we have implemented antiemetic prophylaxis with ondansetron before induction of anaesthesia, on the day of surgery and on the first day after surgery.

Response 8: Metamizole in some cases, is still incorrectly classified as a non-steroidal anti[1]inflammatory drug *Jasiecka, A., Maślanka, T., & Jaroszewski, J. J. (2014). Pharmacological characteristics of metamizole. Polish Journal of Veterinary Sciences, 17(1), 207–214]. In our study metamizole was given as a ‘’rescue drug’’ In Polish guidance is written ‘’Metamizole belongs to the group of non-opioid analgesics. It is a drug with no anti-inflammatory activity, however its metabolites have an inhibitory effect on the synthesis of prostaglandins, mainly on the activity of cyclooxygenases 1 and 2 (COX-1, COX-2). …Metamizole is proven to have synergistic action with NSAIDs, paracetamol and opioid analgesics and constitutes an important component of combined analgesic therapy…..The analgesic strength of a 2.5 g dose of metamizole is comparable to a 10 mg dose of morphine (level II according to EBM)…. In Poland, there were no cases of agranulocytosis after the use of metamizole despite the total consumption of over 110 million tablets per year…’’ *Postoperative pain relief in general surgery - recommendations of the Association of Polish Surgeons, Polish Society of Anaesthesiology and Intensive Therapy, Polish Association for the Study of Pain and Polish Association of Regional Anaesthesia and Pain Treatment. Polski Przeglad Chirurgiczny 2019, 91(1), 47– 68] Single intravenous doses of metamizole in the prevention or treatment of postoperative pain are well tolerated in children. It is commonly used in paediatric patients in some countries, for example in Germany. Due to the risk of agranulocytosis after long-term use metamizole is recommended for short-term postoperative [Fieler M, Eich C, Becke K, et al. Metamizole for postoperative pain therapy in 1177 children: A prospective, multicentre, observational, postauthorisation safety study. Eur J Anaesthesiol. 2015;32(12):839-843].

Response 9: We have added to the subsection 4.1. Strengths and Limitations: ‘’Findings related to patient satisfaction may be limited due to lack of measurement using validated tools’’.

Reviewer 1 Report (Previous Reviewer 1)

Dear authors,

Thank you for addressing the comments raised in the previous review.

Answers to Reviewer 2

Response 1: In the revised Introduction section, we have added: ‘’Multimodal analgesia regimens can utilize varying combinations of opioid, non-opioid analgesics, and regional analgesic techniques [6]. Thoracic epidural, in many institutions, is the standard of perioperative analgesia in children with chest wall deformity [7, 8, 9]. However, intravenous analgesia may also be effective pain management option [10]’’. In the revised Discussion section, we have added: ‘’We used metamizole as a ‚rescue drug‛ due to its strong analgesic effect (the analgesic strength of a 2.5 g dose of metamizole is comparable to a 10 mg dose of morphine; level II according to EBM) and synergistic action with non-steroidal anti-inflammatory drugs, paracetamol and opioid analgesics [29]. The short-term metamizole use for treatment of postoperative pain in children seems to be well tolerated and safe (no clinical signs of agranulocytosis were reported) [30].

Reviewer 2 Report (Previous Reviewer 2)

Thank you for answering my points. I believe the paper is now suitable for publication.

best regards

This manuscript is a resubmission of an earlier submission. The following is a list of the peer review reports and author responses from that submission.

Round 1

Reviewer 1 Report

Dear authors,

Thank you for submitting your work. Please go through my comments.

Using a placebo is the biggest limitation of this study. It would have been fine if authors used an existing standard of care (some other analgesic) to compare with gabapentin.

Why use two primary outcomes? This really questions the sample size used for this study.

The optimal post-operative pain management involves a multimodal approach-line 55. Describe the multimodal approach with appropriate references.

Mention the outcomes at the end of introduction.

Lines 113-114: non-steroidal anti-inflammatory drugs every 8 hours 113 (ketoprofen intravenously – a maximum of 200 mg daily; ibuprofen orally or rectally up 114 to 30 mg/kg daily. The NSAID use does not appear to be standardized here.

Lines 119-121: why so many anti-emetics were in methodology? How was it decided what to you use and when? Was an Apfel score documented prior to surgery?

Lines 126-7: The primary endpoint with respect to efficacy of pain management was postoperative pain intensity and the consumption of morphine- what time postoperative?

Line 178: Assuming a power of 80% and 90% for morphine consumption- postoperative pain is also a primary outcome along with morphine consumption. This sample size is not appropriate for two primary outcomes.

Line 270-3.2.1. Metamizole consumption: above mentioned is ketoprofen and ibuprofen. Now, metamizole is another drug emerging. How?

161-3: Why any validated patient satisfaction score was used in this study?

More in limitations: small sample size, placebo group instead of a standard of care, pediatric patients mainly- subjective assessment issues possible.

Author Response

Response to Reviewer 1 Comments

Dear Reviewer,

We would like to thank you for the detailed review of our manuscript and all of your valuable remarks. We did our best to correct the manuscript accordingly.

With kind regards,

Authors

Point 1: Using a placebo is the biggest limitation of this study. It would have been fine if authors used an existing standard of care (some other analgesic) to compare with gabapentin. More in limitations: small sample size, placebo group instead of a standard of care, pediatric patients mainly- subjective assessment issues possible.

Response 1: The placebo-conrolled trail is regarded as the gold standard for testing the efficacy of new treatment, and can detect treatment effects with a smaller sample size. The placebo-controlled trial measures the total pharmacologically mediated effect of treatment. In contrast, an active control trial measures the effect relative to another treatment. Placebo-controlled trials without active control give little useful information on comparative efficacy [https://www.ema.europa.eu/en/documents/scientific-guideline/ich-e-10-choice-control-group-clinical-trials-step-5_en.pdf].

Therefore, we agree that this was a limitation of this study. As suggested, we have added to subsection 4.1. Strengths and Limitations: ‘’The study was also limited by: small sample size, placebo-controlled study with no active control, subjective nature of pain assessment.’’

Point 2: Why use two primary outcomes? This really questions the sample size used for this study. Line 178: Assuming a power of 80% and 90% for morphine consumption- postoperative pain is also a primary outcome along with morphine consumption. This sample size is not appropriate for two primary outcomes.

Response 2: The minimal sample size was calculated based on pain intensity (before subject enrolment) and morphine consumption (two years after the trial commenced), because we concluded that apart from pain intensity, morphine consumption could also be an important indicator of treatment effectiveness. Therefore, we believe that results of pain score (at least 19 patients in each group are needed to obtain 90% power of difference detection) and morphine consumption (at least 24 patients in each group are needed to obtain 80% power of difference detection) are reliable.

Point 3: Lines 126-7: The primary endpoint with respect to efficacy of pain management was postoperative pain intensity and the consumption of morphine - what time postoperative?

Response 3: This essue was already explained in line 135-140.

Point 4: The optimal post-operative pain management involves a multimodal approach-line 55. Describe the multimodal approach with appropriate references.

Response 4: In the revised Introduction section, we have added ‘’ The optimal post-operative pain management involves a multimodal approach [6]. Multimodal analgesia regimens can utilize varying combinations of opioid, non-opioid analgesics (e.g. paracetamol, nonsteroidal anti-inflammatory drugs, metamizole), analgesic adjuncts (e.g. gabapentin), regional analgesic techniques (e.g thoracic epidural) [7,  8, 9, 10].’’

Point 5: Mention the outcomes at the end of introduction.

Response 5: The purpose of the work includes outcomes.

Point 6: Lines 113-114: non-steroidal anti-inflammatory drugs every 8 hours (ketoprofen intravenously – a maximum of 200 mg daily; ibuprofen orally or rectally up 14 to 30 mg/kg daily. The NSAID use does not appear to be standardized here.

Response 6: The drugs were administered in accordance with the Summary of Product Characteristics in force in Poland. The intravenous ketoprofen formulation was only licensed for patients aged from 15 years. The intravenous ibuprofen was not available when the study was provided (it is licensed for patients aged from 6 years from 2021-09-24). For this reason, two different NSAIDs were used. Ibuprofen was given orally if the patient was able to swallow the drug. Drug were dosed according to body weight, e.g. ketoprofen 1 mg/kg, but in patients weighing > 70 kg the maximum dose was 200 mg daily.

Point 7: Lines 119-121: why so many anti-emetics were in methodology? How was it decided what to you use and when? Was an Apfel score documented prior to surgery?

Response 7: We did not assess the risk of nausea and vomiting for each patient according to Apfel score for adult. Knowing, however, that the postoperative risk of nausea and vomiting according to the Apfel scale is 39-61% (gender, use of postoperative opioids) and moderate or high according to the VPOP score for the pediatric population (age, duration of anesthesia >45 min, and multiple opioid doses), we have implemented antiemetic prophylaxis with ondansetron (0.1 mg/kg up to 4 mg) on the day of surgery and on the first day after surgery. For symptomatic treatment of nausea and vomiting, metoclopramide hydrochloride (0.1-0.2 mg/kg) and/or dexamethasone (0.15 mg/kg up to 5 mg) were additionally administered.

We have updated the information about using antiemetics in 2.3. Interventions section of the revised Materials and Methods.

Point 8: Line 270-3.2.1. Metamizole consumption: above mentioned is ketoprofen and ibuprofen. Now, metamizole is another drug emerging. How?

Response 8: Metamizole is an non-opioid drug used as additional drug (called in the paper as ‘’rescue drug’’ ). This agent is not classified as a non-steroidal anti-inflammatory drug.

Due to the risk of agranulocytosis after long-term use metamizole is recommended for short term postoperative. It is commonly used in paediatric patients in Germany. Single intravenous doses of metamizole in the prevention or treatment of postoperative pain are well tolerated in children [https://doi.org/10.1097/EJA.0000000000000272] and seems to be safe [https://doi.org/10.1371/journal.pone.0122918].

Point 9: 161-3: Why any validated patient satisfaction score was used in this study?

Response 9: We have added to the subsection 4.1. Strengths and Limitations: ‘’Findings related to patient satisfaction may be limited due to lack of measurement using validated tools’’.

Reviewer 2 Report

Dear Authors,

I read with interest your article entitled "The effects of gabapentin on post-operative pain and anxiety, morphine consumption and patient satisfaction in paediatric patients following the Ravitch procedure". The protocol is well designed, and has the merit of being randomized and double-blinded; the article is very well written and very interesting besides clearly exposed and pleasurable to read. 

It would be interesting for me if you could discuss also which kinds of regional anesthesia are potentially useful for the procedure and if they are successful or not, if litereture exists etc; also, why you choose metamizole as adjunct in case of rescue analgesia needed. 

Congratulations!

Author Response

Response to Reviewer 2 Comments

Dear Reviewer,

We would like to thank you for the detailed review of our manuscript and all of your valuable remarks. We did our best to correct the manuscript accordingly.

With kind regards,

Authors

Point 1: It would be interesting for me if you could discuss also which kinds of regional anesthesia are potentially useful for the procedure and if they are successful or not, if litereture exists etc; also, why you choose metamizole as adjunct in case of rescue analgesia needed.

Response 1: In the revised Introduction section, we have added ‘’ The optimal post-operative pain management involves a multimodal approach [6]. Multimodal analgesia regimens can utilize varying combinations of opioid, non-opioid analgesics (e.g. paracetamol, nonsteroidal anti-inflammatory drugs, metamizole), analgesic adjuncts (e.g. gabapentin), regional analgesic techniques (e.g thoracic epidural) [7, 8, 9, 10].’’

Metamizole was used as additional drug. Due to the risk of agranulocytosis after long-term use metamizole is recommended for short term postoperative. It is commonly used in paediatric patients in Germany. Single intravenous doses of metamizole used for the prevention or treatment of postoperative pain are well tolerated in children [https://doi.org/10.1097/EJA.0000000000000272] and seems to be safe [https://doi.org/10.1371/journal.pone.0122918].

Therefore, we have added in the revised Discussion section: ‘’The short-term metamizole use for treatment of postoperative pain are well tolerated in children [8] and seems to be safe [29].’’